# *Scutellaria baicalensis* Induces Cell Apoptosis and Elicits Mesenchymal–Epithelial Transition to Alleviate Metastatic Hepatocellular Carcinoma via Modulating HSP90β

**DOI:** 10.3390/ijms25053073

**Published:** 2024-03-06

**Authors:** Tung-Ho Wu, Tung-Yi Lin, Pei-Ming Yang, Wen-Tai Li, Chau-Ting Yeh, Tai-Long Pan

**Affiliations:** 1Surgical Critical Care Division of Cardiovascular Surgical Department, Kaohsiung Veterans General Hospital, Kaohsiung 813, Taiwan; thwu@isca.vghks.gov.tw; 2Department of Traditional Chinese Medicine, Chang Gung Memorial Hospital at Keelung, Keelung 204, Taiwan; tungyi30@cgmh.org.tw; 3TMU Research Center of Cancer Translational Medicine, Taipei 110, Taiwan; yangpm@tmu.edu.tw; 4Graduate Institute of Cancer Biology and Drug Discovery, College of Medical Science and Technology, Taipei Medical University, Taipei 110, Taiwan; 5National Research Institute of Chinese Medicine, Ministry of Health and Welfare, Taipei 112, Taiwan; lwt0220@nricm.edu.tw; 6Liver Research Center, Chang Gung Memorial Hospital, Taoyuan 333, Taiwan; chautingy@gmail.com; 7School of Traditional Chinese Medicine, Chang Gung University, Taoyuan 333, Taiwan; 8Research Center for Food and Cosmetic Safety and Chinese Herbal Medicine, College of Human Ecology, Chang Gung University of Science and Technology, Taoyuan 333, Taiwan

**Keywords:** hepatocellular carcinoma, *Scutellaria baicalensis*, metastasis, HSP90β, cell cycle, apoptosis

## Abstract

Hepatocellular carcinoma is one of the most common malignant tumors in the world and shows strong metastatic potential. Current medicine for hepatocellular carcinoma therapy is invalid, while *Scutellaria baicalensis* Georgi exhibits the pharmaceutical potential to treat liver diseases and liver cancer. Herein, we verified the inhibitory properties and the pivotal molecules regimented by *Scutellaria baicalensis* on advanced hepatocellular carcinoma. At first, the viability of SK-Hep-1 cells was significantly reduced under treatment of *Scutellaria baicalensis* extract in a dose-dependent manner without affecting the growth of normal hepatocyte. *Scutellaria baicalensis* extract application could remarkably cause apoptosis of SK-Hep-1 cells through p53/cytochrome C/poly-ADP ribose polymerase cascades and arrest the cell cycle at the G1/S phase by downregulating cyclin-dependent kinases. Meanwhile, administration of *Scutellaria baicalensis* extract remarkably attenuated the migration capability as well as suppressed matrix metalloproteinase activity of advanced hepatocellular carcinoma cells. The proteome profiles and network analysis particularly implied that exposure to *Scutellaria baicalensis* extract downregulated the expression of HSP90β, and the clinical stage of hepatocellular carcinoma is also positively correlated with the HSP90β level. Combined treatment of *Scutellaria baicalensis* extract and HSP90β siRNAs could markedly enhance the ubiquitination activity and the degradation of vimentin to subsequently inhibit the metastatic property of SK-Hep-1 cells. Moreover, application of *Scutellaria baicalensis* extract and HSP90β siRNAs depleted phosphorylation of AKT, which stimulated the expression of p53 and consecutively triggered cell apoptosis. These findings suggest that HSP90β may be a prospective target for the effective therapy of advanced hepatocellular carcinoma via accelerating apoptosis of hepatocellular carcinoma cells and eliciting mesenchymal–epithelial transition with the administration of *Scutellaria baicalensis* extract.

## 1. Introduction

Hepatocellular carcinoma (HCC) ranks among the leading causes of cancer-mediated death, and is induced by various risk factors, including virus infection, alcohol abuse, obesity, and metabolic diseases [1,2]. Metastasis and recurrence are the major events associated with poor prognosis and lower survival rates [3]. For the treatment of HCC, several strategies such as surgery, chemotherapy, radiation therapy, immunotherapy, and targeted therapy have been applied, but most of these administrations show little success in the advanced stage [4]. Therefore, alternative medicine and the related mechanisms need to be revealed urgently for effective therapeutics in HCC metastasis [5,6,7]. In the current study, systematic analysis was utilized to reveal the global changes in proteins and molecules for a comprehensive exploration of cellular interactions and molecular pathways involved in the inhibitory efficacy of Chinese medicine on HCC progression.

Targeting a single factor of a signaling pathway may be ineffective due to the complexity of the biological process in HCC [8]. Traditional Chinese medicine is characterized by being multi-ingredient, multi-target, and showing low adverse effects for the therapy of hepatic disorders [9,10,11]. Especially, *Scutellaria baicalensis* (*S. baicalensis*) has been employed for relief from fever, detoxification, dehumidification, and immune regulation [12]. It is also a major ingredient of herbal formulae widely prescribed in Asian countries for the treatment of hepatic diseases [13]. The bioactive components of *S. baicalensis*, including baicalein, baicalin, wogonin, and wogonoside, have been identified to exhibit anticancer, antiviral, and anti-inflammatory properties [14]. Previous reports have indicated that *Scutellaria baicalensis* extract (SBE) causes apoptosis in HSC-T6 cells by activating extracellular signal-regulated kinase (ERK)-p53 cascades and enhancing the caspase system [15]. However, the detailed mechanisms by which SBE alleviates advanced liver cancer are necessary for controlling HCC deterioration and novel drug development.

Heat shock proteins (HSPs) are cellular chaperone proteins that function in diverse physiological and protective processes to assist in maintaining cellular homeostasis. In particular, HSP90 facilitates cell cycle control, protein folding/assembly as well as signal transduction pathways and is highly increased under environmental stresses [16]. Furthermore, HSP90 serves as a molecular chaperone for various client proteins including receptor tyrosine kinases, metabolic enzymes, and epigenetic regulators that are essential for the proliferation and sustenance of cancer cells [17]. A disturbance in the interaction between HSP90 and its client proteins leads to their destabilization and degradation via the proteasome-mediated pathway, eventually activating caspase pathways and cell apoptosis [18]. As a result, HSP90 is an important factor associated with tumorigenicity, therapy resistance, and inhibiting apoptosis [19,20]. Most importantly, although HSP90 is highly expressed in various malignancies [21], its role in HCC remains unclear.

Chinese medicine may regulate the progression of HCC, which would result in great changes of proteins and molecules in quantity and quality. Therefore, high throughput technologies combined with data mining using bioinformatics tools have offered a benefit for large-scale screening and identifying differential expression profiles that reflect the therapeutic effects of *S. baicalensis*. The 2-dimensional electrophoresis (2-DE) analysis combined with matrix-assisted laser desorption ionization-time of flight mass spectrometry (MALDI-TOF MS) and the “signature network” analysis using MetaCore™ software (version 5.2) reveal the molecular mechanisms and signaling pathways based on their biological functionality and integration of molecular information [22,23,24].

Collectively, our findings will help to develop novel therapeutic strategies for minimizing the risk of HCC metastasis and also benefit chemotherapy in advanced HCC.

## 2. Results

### 2.1. Inhibitory Effect and Cytotoxicity of SBE upon SK-Hep-1 and FL83B Cells

FL83B and SK-Hep-1 cells were exposed to 0~1000 μg/mL SBE for 24 h and cell viability was determined by in vitro MTT assays to verify the pharmaceutical effects of SBE. The growth of SK-Hep-1 cells was significantly inhibited under SBE application in a dose-dependent manner (Figure 1A), whereas the viability of FL83B cells was unaffected, suggesting that SBE could effectively suppress the viability of malignant SK-Hep-1 cells without causing damage to normal hepatocytes. Next, assessment of the inhibitory effect of SBE on the cell cycle of SK-Hep-1 cells by flow cytometric analysis revealed apoptosis in only a small proportion of the control group of SK-Hep-1 cells; however, the cells shifted remarkably toward the sub-G1 phase with ratios of 11.68% and 27.76%, respectively, when exposed to 250 and 500 μg/mL SBE for 24 h, implying that SBE arrested SK-Hep-1 cells at the G1/S phase and consequently induced apoptosis (Figure 1B).

### 2.2. SBE Activated the Key Molecules of the Apoptotic Pathway in the SK-Hep-1 Cell

To investigate if this pro-apoptotic effect of SBE is induced by activated forms of key proteins in the apoptotic pathway, we evaluated the levels of caspase-3, caspase-7, caspase-9, and poly-ADP ribose polymerase (PARP) by Western blotting. Cleaved forms of caspase-3 (19 kDa), caspase-7 (20 kDa), caspase-9 (37 kDa), and PARP (89 kDa) were significantly increased under SBE treatment in a dose-dependent manner (Figure 2A). Furthermore, the administration of a pan-caspase inhibitor (z-VAD-FMK) could significantly abolish SBE-mediated apoptosis, which manifested as attenuated cleavage of caspase-3, caspase-7, caspase-9, and PARP (Figure 2B). Tumor suppressor protein p53 is an important regulation factor of apoptosis; therefore, we then surveyed whether p53 also contributes to SBE-induced apoptosis. Bcl-2-associated X (Bax) protein is another pro-apoptotic factor, which increases apoptosis in cells while the B-cell lymphoma 2 (Bcl-2) protein could inhibit programmed cell death and corresponds with the p53 status of the cells. Compared with the control non-SBE-treated cells, those treated with SBE exhibited a dose-dependent decrease in the levels of Bcl-2 expression whereas the levels of p53 and pro-apoptotic Bax protein were correspondingly enhanced with increasing doses of SBE. The ratio of Bax/Bcl-2 protein expression under SBE exposure increased dose-dependently, suggesting the susceptibility of SK-Hep-1 cells toward apoptosis.

Cellular apoptosis is caused by the disruption of mitochondrial function through the release of cytochrome c from mitochondria into the cytosol, thus stimulating caspases in response to diverse inducers of cell death, eventually leading to apoptosis. We further determined the levels of cytochrome c after SBE treatment. Again, cytoplasmic cytochrome c increased dramatically following administration of SBE in a dose-dependent manner (Figure 2C). As we know, the induction of apoptosis is controlled by a series of cell cycle regulatory proteins, including cyclins and cyclin-dependent kinases (CDKs). Therefore, we further examined the efficacy of SBE in inhibiting cancer cell proliferation and inducing cell apoptosis by regulating the cyclins and CDKs. As indicated in Figure 2D, SBE treatment caused cell cycle arrest at the G1/S phase as well as distinctly downregulating the expression of CDK2, CDK4, CDK6, cyclin D, and cyclin E in a dose-dependent manner. These results indicated the molecular mechanism by which SBE suppresses CDK4/CDK6 to exhibit its anti-cancer activity.

### 2.3. SBE Mitigated Epithelial-Mesenchymal Transition (EMT), Migration, and Matrix Metalloproteinase (MMP) Activity of SK-Hep-1 Cells

EMT is a cellular program that confers on cancer cells increased tumor-initiating and metastatic potential. Upon activation of EMT, E-cadherin expression is repressed while neural cadherin (N-cadherin) and vimentin would be induced. Western blot analysis verified that exposure of SK-Hep-1 cells to SBE significantly increased the expression of E-cadherin relative to the control, whereas the levels of N-cadherin and vimentin were significantly downregulated, suggesting that SBE could reverse the EMT of advanced HCC cells into mesenchymal–epithelial transition (MET). Additionally, low claudin expression seems to correlate with poor differentiation and poor prognosis in HCC. As expected, SBE treatment could obviously induce the expression levels of claudin, implying the role of SBE in diminishing EMT (Figure 3A). According to our findings, 250 μg/mL SBE could effectively suppress the growth of SK-Hep-1 cells; we used 250 μg/mL SBE as the working dosage for further experiments. Consistently, scratch-wound assays demonstrated that 250 μg/mL SBE application significantly attenuated SK-Hep-1 cell migration compared with the control group and the gap was not filled by cells within 72 h (Figure 3B). Further analysis of the anti-metastatic effects of SBE on SK-Hep-1 cells using gelatin-zymography revealed obviously inhibited MMP activity in SBE-treated cells in a dose-dependent manner compared to the control sample, demonstrating that treatment of SBE could significantly alleviate the invasive potential of advanced HCC cells (Figure 3C). These results suggested that SBE could effectively suppress tumor metastasis.

### 2.4. Characterization of Differential Protein Profiles and Functional Network Analysis

Two-dimensional electrophoresis, along with MALDI-TOF MS, is a powerful tool to delineate differential protein expression levels with high sensitivity and efficacy. The protein extracts from SK-Hep-1 cells treated with or without 250 μg/mL SBE were separated by 2-DE. In total, 1028 protein spots were observed in the 2-DE maps, and protein spots of interest were further identified by MALDI-TOF. In particular, 25 proteins potentially related to HCC progression were noted (Figure 4A); the comprehensive results of the spectrometric analyses and protein functions are summarized in Table 1. Western blotting was performed to verify the protein expression derived from the 2-DE profiles. Significant decrease in levels of HSP90β, HSP70, and α-tubulin were noted in the SBE-treated group compared to the control, whereas HSP60 showed enhanced expression consistent with the changes in those proteins in proteome experiments (Figure 4B). Then, the MetaCore™ analytical tool was applied to further delineate the relationship of differentially expressed proteins discovered in the proteomics analysis and the significance of the mechanisms by which SBE administration could alleviate the deterioration of HCC. This algorithm builds biological networks from uploaded proteins and assigns a biological process to each network. As shown in Figure 4C, protein–protein interaction networks using the shortest-path algorithm demonstrated that differentially expressed proteins in the presence of SBE were closely linked to the statistically significant networks as follows: protein folding (8.069 × 10^−09^), cytoskeletal intermediate filaments (5.569 × 10^−07^), cell cycle (2.738 × 10^−06^), cell adhesion (3.270 × 10^−05^), and immune response (8.436 × 10^−05^). According to the network analysis, SBE could regulate the key protein HSP90β, subsequently influencing the expression of vimentin and disturbed cytoskeleton arrangement. Dysregulation of HSP90β under SBE treatment also affected cell-cycle-modulated proteins, finally leading to cell apoptosis. These findings explain the involvement of the inhibitory efficacy of SBE on HCC cells and the targeted proteins (Figure 4D).

### 2.5. Significance of HSP90β in Controlling Metastatic Hepatocellular Carcinoma under SBE Administration

To explore the functional roles of HSP90β in different stages of HCC cells, in-situ expression of HSP90β was identified in the clinical samples. HSP90β was highly expressed in the cytosolic and nuclear region of tumor and peritumoral liver tissue of stage 2 or 3 HCC samples, compared to that in the normal tissue and lower grade of HCC (Figure 5A). We further quantified the expression level of HSP90β in the HCC tissue array and indicated that significantly increased levels of HSP90β presented in patients at late stage and metastatic samples while slightly higher levels of HSP90β were showed in the subjects of early stage compared with the control group (Appendix A), suggesting that HSP90β may be a promising biomarker in detecting advanced HCC. Accordingly, synergistic application of 250 μg/mL SBE and HSP90β siRNAs could obviously suppress the levels of vimentin and N-cadherin compared to those in the control or mock groups (Figure 5B). Meanwhile, increased ubiquitination contributing to significant degradation of vimentin was identified as a result of the combined administration of SBE and HSP90β siRNAs with respect to the individual treatment (Figure 5C). Protein kinase B (AKT) has been reported to modulate some cell-cycle events; thus, we next determined whether concomitant exposure of SBE and HSP90β siRNAs could regulate phosphorylated (p)-AKT and associated cell-cycle proteins to induce apoptosis. As expected, synergistic treatment of SBE and HSP90β siRNAs could significantly suppress AKT phosphorylation, which induces p53-dependent gene transactivation and reduces CDK2 as well as p21 expressions (Figure 5D), indicating that apoptotic induction and cell cycle arrest are closely related to the inhibition of HSP90β in SK-Hep-1 cells.

## 3. Discussion

HCC is the third leading cause of cancer-related death and its occurrence is still rising worldwide. Metastasis and recurrence are the major reasons related to poor prognosis or lower post-surgery survival rate in HCC patients [25]. The majority of patients who are diagnosed at an advanced stage have finite treatment options and the prognosis of HCC remains quite poor while *S. baicalensis* has been widely utilized for curing various hepatic disorders in Asia [26,27,28]. Although some flavone components have been identified in the root tissue of *S. baicalensis*, the functional roles of multiple component mixtures extracted from *S. baicalensis* in treating advanced HCC have not been fully elucidated. We investigated the inhibitory effects and identified the molecular targets of SBE against the SK-Hep-1 cells with the highly metastatic potential to treat and further prevent HCC progression.

MTT analysis showed that SBE could significantly alleviate SK-Hep-1 growth (IC_50_ 524.74 μg/mL) while the cytotoxicity of SBE on FL83B cells (IC_50_ 3580.44 μg/mL) is relatively mild, suggesting that SBE could selectively suppress advanced HCC in a dose-dependent manner without damaging normal liver cells. Accordingly, apoptosis is suggested as an effective process to eliminate cancer cells; therefore, we found that SBE could arrest the growth of SK-Hep-1 cells in the G1/S phase with concomitant cleavage of PARP and caspase-3, -7, and -9 at 24 h and the following increase in cell number in their apoptosis phase. Pretreatment with a specific pan-caspase inhibitor markedly prevented the cleavage of PARP. Our results indicated that SBE could promote the expression of p53, leading to upregulated expression of the proapoptotic Bax protein. We also demonstrated that SBE administration inhibited Bcl-2 expression and induced cytochrome c release, an irreversible step toward apoptosis. Our results indicate that treatment of SBE inhibited SK-Hep-1 cell proliferation and promoted cell apoptosis by inducing mitochondrial cytochrome c release, caspase activation, and cell cycle arrest.

Furthermore, apoptosis and cell cycle regulation depend on the precise modulation of specific cyclins associated with CDKs [29]; cyclins D, E, and CDK2 enhance cancer cell proliferation by modulating the G0/G1 to the S phase transitions in tumor cells. In this regard, high levels of cyclins D, E, and CDK2 are also closely linked to poor prognosis in HCC [30]. Herein, SBE application resulted in the downregulation of cyclin D, E, and CDK2/4/6, which hindered the progression of HCC cells from the G1 phase to the S phase, as was observed in the flow cytometry analysis. Altogether, SBE application could suppress the G1/S cell cycle transition and proliferation of SK-Hep-1 cells.

Our previous study indicated that vimentin may be a promising candidate in regulating HCC metastasis [31]. Herein, SK-Hep-1 cells treated with SBE showed reduced vimentin and N-cadherin expression, whereas E-cadherin and claudin were increased, demonstrating the reversion in EMT of SK-Hep-1 cells under SBE exposure. On the other hand, MMPs play crucial roles in tumor invasion and angiogenesis by mediating the degradation of the ECM of cancer cells, and we also observed that MMP activity decreased significantly after treatment of SBE in a dose-dependent manner. In addition, SBE also exhibited inhibition of cell motility. These findings support the potential of SBE as a feasible chemotherapeutic agent against advanced HCC via alleviating EMT initiation or maintenance [32].

Functional proteomics analysis identified differential protein expression and explored mechanisms by which SBE affects HCC cell progression. Our investigations revealed 25 significantly altered protein spots with or without 250 μg/mL SBE treatment. These proteins were grouped into several categories according to their known functions, including protein folding, cytoskeleton regulation, and cell cycle. The most striking feature is that HSPs, which assist in protein folding and stabilize proteins, showed significant changes in quality and quantity under different treatments. HSPs are highly conserved proteins and are remarkably induced in response to cellular stresses. Therefore, overexpressed HSPs play a crucial role in tumorigenesis, metastasis, and therapeutic resistance [33,34,35]. Using specific inhibitors to target HSPs provides an attractive therapeutic strategy for the diagnosis, prognosis, and improvement of outcomes for HCC patients. Network analysis revealed that HSP90β in particular plays a central role in modulating the EMT, cell cycle, migration, and apoptosis, as mentioned in the previous studies, which suggest that HSP90β could promote endothelial cell-dependent tumor angiogenesis, resulting in HCC occurrence and development [36,37,38]. Consistently, enhanced expression of HSP90β is linked with advanced stages and inappropriate survival of the HCC, highlighting the roles of HSP90β in indicative EMT events in clinical specimens. Thus, expression levels of HSP90β can function as a biological molecule in predicting HCC development. Some studies have shown that the elevated expression of nuclear HSP90 could be observed in several cancers, such as breast cancer and non-small cell lung cancer, and nuclear accumulation of HSP90 might indicate the increase of tumor cell invasion as well as poor survival of cancer patients [39,40]. As expected, nuclear HSP90β over-production is also shown in the late stages of HCC subjects, suggesting that HSP90β plays a pivotal physiological function in chromatin remodeling and contributes to the development of HCC. Proteomic and Western blot analyses verified that SBE administration caused significant reduction of HSP90β expression. Furthermore, combined administration of SBE with HSP90β siRNAs showed synergistic effects in the diminishment of vimentin by activating ubiquitination, which triggers the changes in cell shape, adhesion, and motility during cancer progression. Hence, stalling the expression of HSP90β may disturb the expression and stability of vimentin through the ubiquitin system, resulting in impaired adhesion and motility of metastatic HCC cells.

Blocking AKT phosphorylation has often been shown to alleviate cell cycle disorder and malignant proliferation of cancer cells [41]. The activation of the AKT signaling pathway is associated with poor prognosis in HCC patients [42]. The binding between AKT and HSP90β could stabilize the structure to assist AKT phosphorylation. In our study, combined treatment with SBE and HSP90β siRNAs resulted in decreased phosphorylation of AKT at Ser473, which prevents cells from apoptosis, finally suppressing the expression levels of CDK2, a key S-phase regulatory factor, and p21 to arrest SK-Hep-1 cells in the G1/S stage. These findings imply that targeting HSP90β might be as effective as modulating specific signaling pathways for advanced HCC therapy under add-on treatment of SBE. We suggest that SBE and HSP90β siRNAs exert synergistic effects in various events to attenuate malignant HCC, demonstrating that SBE should influence multiple targets, except HSP90β, which might be of major advantage in abrogating metastatic potential in HCC cells.

Tumor development is also closely associated with some emerging issues such as the RNA molecules and microbiota [43,44]. Some critical pathways related to the PI3K/AKT/mTOR cascades, EMT as well as the cell cycle are regulated by RNAs including miRNAs and noncoding RNAs (ncRNAs), eventually influencing cancer progression. In addition, gut microbiota can interact with the host to affect various targets and modulate HCC development. Previous reports have suggested that the bioactive components derived from *S. baicalensis* might regulate the release of RNAs and maintain homeostasis of the intestinal microbiota, which plays a significant curative role in HCC [45,46]. We will further investigate the connection among SBE, HSP90β, and HCC development under the regulation of miRNAs or microbiota.

## 4. Materials and Methods

### 4.1. Materials

Specific antibodies to p53, HSP70, HSP60, HSP90β, β-tubulin, p21, GAPDH, and HSP90β siRNAs were purchased from Santa Cruz (Santa Cruz, CA, USA). Polyclonal antibodies to Bax, Bcl-2, cytochrome c, and β-actin were obtained from Millipore (Temecula, CA, USA). Polyclonal antibodies to caspase-3, -7, -9, PARP, CDK2, -4, -6, cyclin D, cyclin E, N-cadherin, E-cadherin, claudin, AKT, and phospho-AKT^Ser473^ were purchased from Cell Signaling (Beverly, MA, USA). Pan-caspase inhibitor (Z-VAD-FMK) was obtained from Enzo life science (New York, NY, USA).

The overall structure flowchart is shown in Figure 6 as follows:

### 4.2. Cell Culture and MTT Assay

The SK-Hep-1 cells were maintained in DMEM medium containing non-essential amino acid, 10% fetal bovine serum (FBS) at 37 °C in a humidified atmosphere of 5% CO_2_. FL83B cells were cultured with F12K medium containing 10%. Cell viability was determined by MTT [47]. A total of 1 × 10^5^ cells were seeded in 24-well plates for 24 h (h) and made quiescent by incubating in medium containing 0.2% FBS overnight. After treating with various concentrations (0, 15.625, 31.25, 62.5, 125, 250, 500, and 1000 μg/mL) of SBE for 24 h, isopropanol solution mixed with tetrazolium salt was added to the wells and incubated for additional 4 h at 37 °C. The optical density of the dissolved material was measured spectrophotometrically at 570 nm, and assays were performed in triplicate. We conducted HPLC analysis for the current batch of SBE and the major components were present in Appendix A.

### 4.3. Flow Cytometric Analysis

The effects of SBE on cell cycle were evaluated by flow cytometry [48]. SK-Hep-1 cells were treated with SBE with the indicated concentrations for 24 h and harvested by centrifugation at 750× *g* for 5 min. Cell pellets were then fixed with ice-cold ethanol for 1 h. Fixed cells were washed with PBS and resuspended in a staining solution containing propidium iodide (PI, 50 μg/mL) and DNase-free RNase (100 μg/mL). Cell suspensions were incubated at 37 °C for 1 h in the absence of light and analyzed on a fluorescence-activated cell sorter flow cytometer (FACScaliber, Becton Dickinson, Franklin Lakes, NJ, USA).

### 4.4. Western Blot Analysis

The protein was isolated using cell lysis buffer (Cell Signaling, Beverly, MA, USA) and the concentrations were measured using the Bradford Protein Assay Kit (AMRESCO, Radnor, PA, USA). Protein lysates were prepared and Western blot analyses were performed as described previously [48]. For ubiquitination assay, cells were washed three times and then lysed with lysis buffer for 30 min. The cell lysate was centrifuged for 10 min at 12,000 rpm and the rest of homogenates were incubated with ubiquitin antibodies overnight at 4 °C. Protein A/G plus agarose beads (Santa Cruz) were added for another 2 h. The immunoprecipitation beads were washed with cold PBS five times, followed by Western blotting analysis. The levels of expression of β-actin and GAPDH were used as a gel loading control. Densitometric analyses of scanned immunoblotting images were performed using GeneTools software (Syngene version 6.05, Cambridge, UK). Results shown are representative of three separate experiments.

### 4.5. Wound-Migration Assay

SK-Hep-1 cells treated with or without 250 μg/mL SBE were grown in 35-mm culture dishes to 80% confluence. A wound was formed using a 200 μL pipette tip to clear the cell monolayer, and the boundary of the wound was marked. Next, cells were incubated for 24 h and cell migration was measured by counting the number of cells that migrated into the clear space using an Olympus microscope (IX71) at 20× fitted with an ocular grid. Values presented are the mean of four different random fields of wounds sampled from three independent experiments. The areas of cell migration were determined by dividing the mean number of cells that moved from the edge to the wounded area in experimental culture by cells that moved from the edge in the control culture.

### 4.6. Gelatin Zymography

Media with or without SBE application were collected, and gelatin zymography was performed as previously described [49]. After electrophoresis, gels were washed twice with 50 mM Tris–HCl, at pH 7.5, containing 2.5% Triton X-100 (*v*/*v*), then incubated at 37 °C in 50 mM Tris–HCl buffer containing 5 mM CaCl_2_ overnight. Digestion was terminated, and gels were stained with 0.5% Coomassie brilliant blue R250 followed by destaining with 10% acetic acid and 10% methanol. Enzyme-digested regions were observed as white bands against a blue background. Zones of enzymatic activity were seen as negatively stained bands.

### 4.7. Two-Dimensional Electrophoresis (2-DE) and Image Analysis

Two-dimensionl electrophoresis is a powerful tool to analyze differential protein expression levels with or without 250 μg/mL SBE treatment according to previously described procedures [50] which have been both high sensitivity and efficacy. The cell lysate (180 μg) was applied to 18-cm IPG, pH 4–7 linear strip (GE Healthcare, Uppsala, Sweden) and separated on the IPGphor III System for the first dimension. The 2-DE was carried out on 10% acrylamide gels (Bio-Rad, Hercules, CA, USA) and the gels were visualized by silver staining. Protein spots were quantified using the Prodigy SameSpots software (Nonlinear Dynamics version 1, Newcastle, UK). More than 1.5-fold alterations at 95% confidence interval (*p* < 0.05) were considered statistically significant. All experiments were repeated three times to confirm reproducibility.

### 4.8. In-Gel Digestion of Proteins and Mass Spectrometric Analysis

Spots of interest were excised and digested in-gel with trypsin for further protein identification according to previously described procedures [51]. After digestion, the tryptic peptides were acidified with 0.5% TFA and loaded onto an MTP AnchorChip^TM^ 600/384 TF (Bruker-Daltonik, Bremen, Germany). MS analysis was performed on an Ultraflex^TM^ MALDI-TOF mass spectrometer (Bruker-Daltonik). Monoisotopic peptide masses were assigned and used for database searches with the MASCOT search engine (Matrix Science, London, UK). Search parameters were set as follows: a maximum allowed peptide mass error of 50 ppm, and consideration of one incomplete cleavage per peptide.

### 4.9. Biological Network Analysis Using MetaCore^TM^

To further explore the relationship of differentially expressed proteins revealed by the 2-DE analysis and their significance in the mechanisms linked to HCC progression, we applied MetaCore™ software (vers. 5.2 build 17389, GeneGo, St. Joseph, MI, USA) to reveal associated ontological classes and relevant pathways. The algorithm builds biological networks from uploaded proteins and assigns a biological process to each network [51].

### 4.10. Clinical Tissue Array

Commercial tissue array was utilized to validate the expression level and pattern of HSP90β in clinical samples. One tissue column (2.0 mm each in diameter) was obtained from each selected paraffin block and arranged in separate new paraffin blocks with 60 holes by using a trephine apparatus (SuperBioChips Laboratories, Seoul, Republic of Korea). These microarray blocks were sectioned at 4 μm thickness and processed for immunohistochemical staining. The sections were rehydrated with graded ethanol and immersed in Tris-buffered saline after removal of paraffin with xylene [52]. IHC with an HSP90β antibody (Santa Cruz, 1:200-diluted in phosphate-buffered saline (PBS)) was applied. Next, sections were counterstained with Mayer’s hematoxylin for 2 min, and slides were evaluated under a light microscope (Olympus BX51, Tokyo, Japan). Digital photomicrographs were then processed with DP-72 (Olympus). The liver tissues were graded with respect to HSP90β expression level as follows: 0, no expression; +1; 0~15%, sporadic expression in a few cells; +2, up to 25% positive cells; +3; 26–50% positive cells; +4, 51–75% positive cells; +5, 76–100% positive cells [53].

### 4.11. Gene Silencing by Small Interfering RNA

In the study, knockdown of HSP90β was achieved by small interfering RNA to evaluate the function of HSP90β in modulation of proteins or molecules associated with SK-Hep-1. Cells were plated onto 6-well plates (1 × 10^5^ cells/well), maintained in antibiotic-free medium for 24 h, and transfected with a mixture containing Opti-MEM, 8 μL/well Lipofectamine 2000 (Invitrogen, San Diego, CA, USA), and either 0.5 μg/well scrambled siRNA (mock) or a mixture of HSP90β siRNAs for 6 h [54]. The sequences of these siRNAs are available from the manufacturer. At 48 h post-transfection, cells were treated with or without 250 μg/mL SBE for 24 h and harvested for following Western blot analyses. All experiments were performed in biological triplicate to confirm reproducibility.

### 4.12. Statistical Analysis

All values were presented as the mean ± standard deviation (SD) of a minimum of 3 replicate tests obtained from three individual experiments. A Kruskal–Wallis (nonparametric ANOVA) test was carried out using SPSS software (SPSS version 17, Chicago, IL, USA) when multiple comparisons were made. Differences were considered significant at * *p* < 0.05, ** *p* < 0.01 and *** *p* < 0.001.

## 5. Conclusions

The current study provides evidence that the molecular pathways underlying SBE-mediated alleviation of advanced HCC are listed as follows: (a) The application of SBE could simultaneously modulate HSP90β levels, thus inducing the degradation of vimentin to abolish the EMT of SK-Hep-1 cells and promoting apoptosis to prevent the progression of HCC. Meanwhile, abrogation of HSP90β due to SBE treatment inhibited AKT phosphorylation and, in turn, suppressed the expression of cyclin D, E, and CDK2/4/6. (b) Over-expression of p53 resulted in Bax/Bcl-2 translocation and cytochrome c release, subsequently leading to caspase-3-7-9-dependent apoptosis under SBE exposure. (c) Additionally, SBE administration also inhibited MMP activity and arrested the migration/invasion of SK-Hep-1 cells (Figure 7).

## Figures and Tables

**Figure 1 ijms-25-03073-f001:**
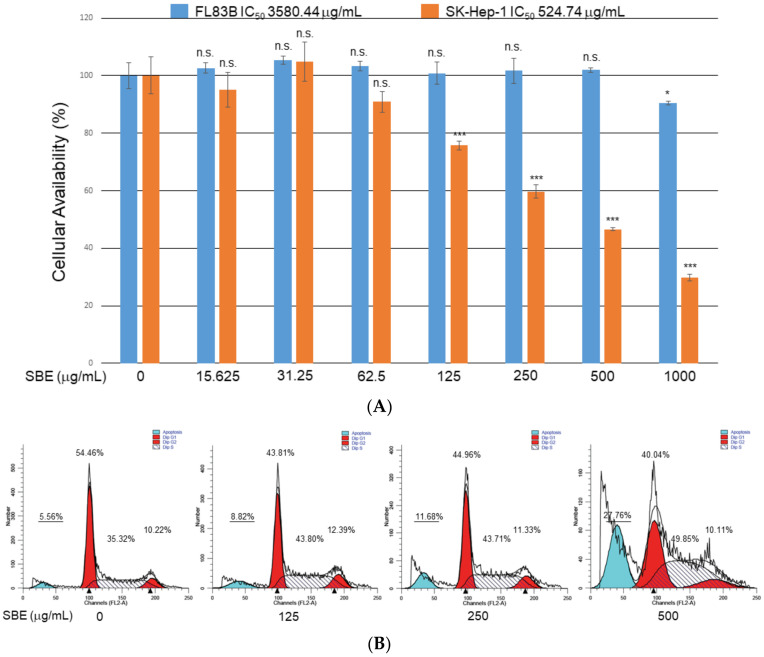
(**A**) Inhibitory effect of SBE at various concentrations upon FL83B and SK-Hep-1 cell viability as measured by MTT assays. FL83B cells were represented by blue columns, and SK-Hep-1 cells were presented in orange columns. Data are means ± SD of three independent experiments. The statistical results were compared with the control groups (* *p* < 0.05, *** *p* < 0.001, n.s. indicated no significance). (**B**) Sk-Hep-1 cells were treated with 0, 125, 250, and 500 μg/mL SBE and analyzed by flow cytometry. The peak of the apoptotic phase characterized by the increase in the sub-G1 cell fraction (blue color) was observed under treatment of 500 μg/mL SBE while G1 and G2 phases were shown by red color; S phase was indicated by slash area. Ratios of cells in the various phases are represented as a percentage.

**Figure 2 ijms-25-03073-f002:**
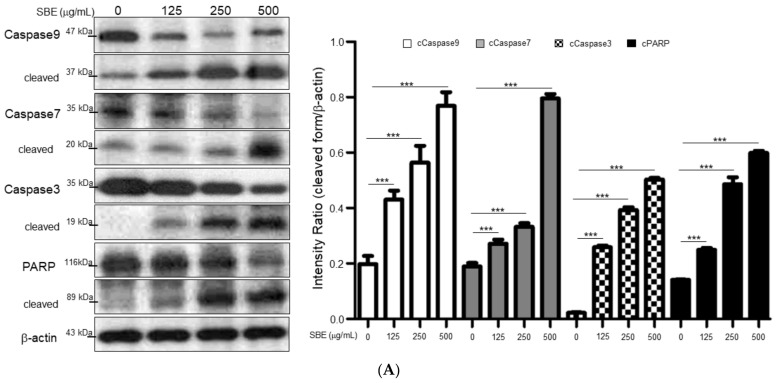
(**A**) Treatment of SBE significantly induced caspase-3, -7, -9 activation and PARP cleavage at 24 h. β-actin was used as an internal control. The quantitative results were demonstrated as bar charts (*** *p* < 0.001). (**B**) The protein levels of cleaved caspase-3, -7, -9, and PARP with or without treatments of Z-VAD-FMK and SBE were determined by Western blotting assays. β-actin was used as an internal control and the quantified results were indicated by the bar chart (* *p* < 0.05, *** *p* < 0.001). (**C**) Western blot analysis for protein levels of Bax, Bcl-2, cytochrome c and p53 with treatment of 0, 125, 250, and 500 μg/mL SBE. Results represent the mean ± SD of three independent experiments (* *p* < 0.05, *** *p* < 0.001). (**D**) Cell cycle arrest at G1/S phase for SK-Hep-1 cells was determined for 24 h by flow cytometric analysis. (**E**) The protein expression of cyclin-CDK including CDK2, CDK4, CDK6, cyclin D, and cyclin E in SK-Hep-1 cells exposed to 0, 125, 250, and 500 μg/mL SBE. GAPDH was applied as the loading control and bar graphs showed the quantitative analysis of the protein expression levels. Results are shown as the mean ± SD (* *p* < 0.05, *** *p* < 0.001, n.s. indicated no significance).

**Figure 3 ijms-25-03073-f003:**
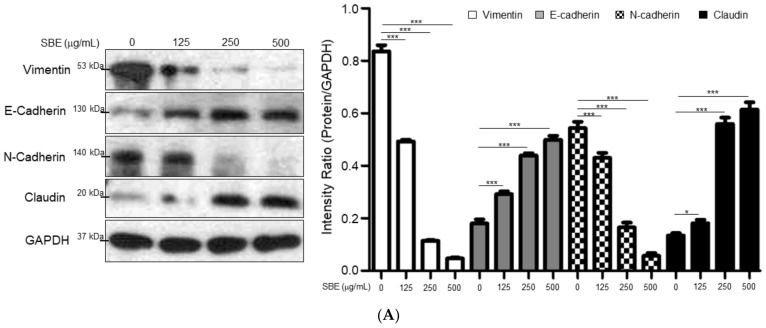
(**A**) Western blotting was used to determine expressions of EMT associated proteins. The expression of vimentin and N-cadherin was remarkably downregulated while the levels of E-cadherin and claudin were obviously increased under application of SBE. GAPDH was used as an internal control, and the relative expression to GAPDH is represented by the bar graph. Data are means ± SD of three independent experiments (* *p* < 0.05, *** *p* < 0.001). (**B**) SBE administration at 250 μg/mL inhibited wound closure. Representative phase-contrast micrographs of scratch-wounded confluent cultures were recorded at 0, 24, 48, and 72 h post-wounding (*** *p* < 0.001). (**C**) A gelatin zymographic assay was performed using SK-Hep-1 cells treated with 0, 125, 250, and 500 μg/mL SBE. The quantitative results are shown as bar charts (*** *p* < 0.001).

**Figure 4 ijms-25-03073-f004:**
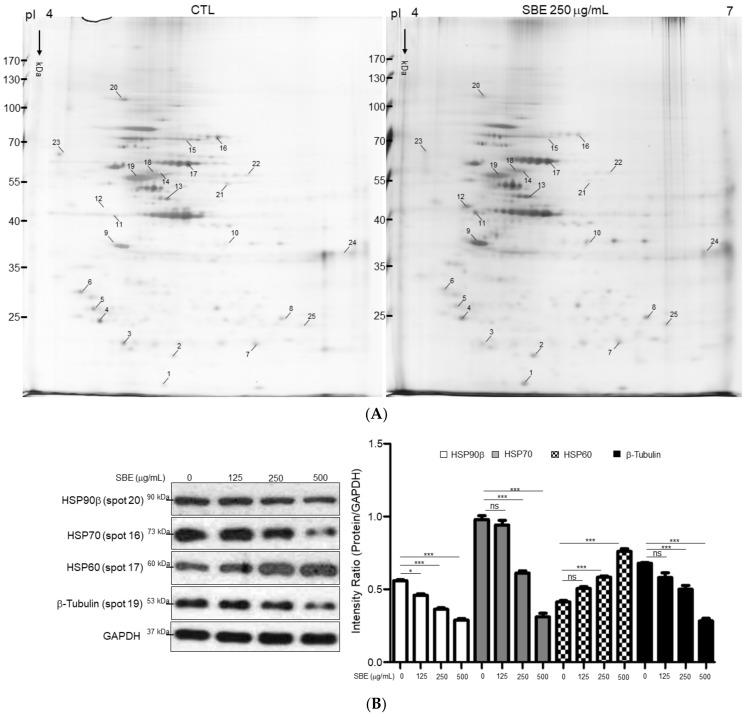
(**A**) Typical two-dimensional gel electrophoresis (2-DE) pattern of SK-Hep-1 cells treated with 250 μg/mL SBE. The protein lysate was focused on a pH 4-7 linear immobilized pH gradient (IPG) strip before being separated on a 10% polyacrylamide gel. Protein spots with significant changes in intensity are labeled with Arabic numerals. (**B**) Western blot analysis was applied to validate protein changes revealed by 2-DE analysis. GAPDH was used as an internal control, and the relative expression to GAPDH is represented by the bar graph. Data are means ± SD of three independent experiments, carried out in triplicate (* *p* < 0.05, *** *p* < 0.001, n.s. indicated no significance). (**C**) Top-ranked pathways from the GeneGo MetaCore™ pathway analysis are indicated. Pathways were ranked according to *p* values, and bars represent the inverse log of the *p* values. The biological processes in this network are mainly involved in protein folding, cell cycle control, and cytoskeletal rearrangement. (**D**) Biological network analyses of differentially expressed proteins using MetaCore™ mapping tools. Nodes represent proteins and lines between the nodes indicate direct protein–protein interactions. The various proteins on the map are indicated by different symbols representing the functional class of the proteins.

**Figure 5 ijms-25-03073-f005:**
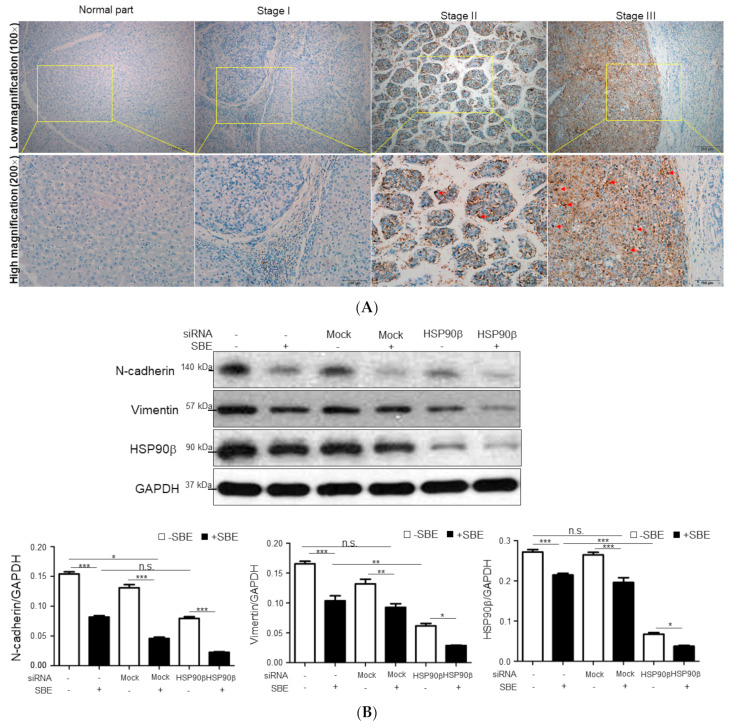
(**A**) Immunohistochemical study of HSP90β expression in adjacent normal part, stage I, and stage III of HCC tissues. The positive signal is presented in brown (bar scale 200 μm) and nuclear HSP90β is indicated by red arrows. (**B**) SK-Hep-1 cells were treated with or without 250 μg/mL SBE and siRNAs of HSP90β. Levels of EMT related protein such as N-cadherin and vimentin as well as HSP90β were measured by Western blot analysis. GAPDH was used as an internal control. The quantified results were indicated by the bar chart (* *p* < 0.05, ** *p* < 0.01, *** *p* < 0.001, n.s. indicated no significance). Results represent the means ± SD of three independent experiments. (**C**) SK-Hep-1 cells were treated with or without 250 μg/mL SBE and siRNAs of HSP90β modulated vimentin ubiquitination. Proteins were isolated from cells and the levels of ubiquitination as well as ubiquitylated vimentin were detected by Western blotting (*** *p* < 0.001). (**D**) Combined application of 250 μg/mL SBE and siRNAs of HSP90β inhibits activation of the AKT signaling pathway. The expression levels of AKT, p-AKT ^Ser473^, p53, CDK2, and p21 in total cell lysates were detected by Western blotting (* *p* < 0.05, ** *p* < 0.01 and *** *p* < 0.001, n.s. indicated no significance).

**Figure 6 ijms-25-03073-f006:**
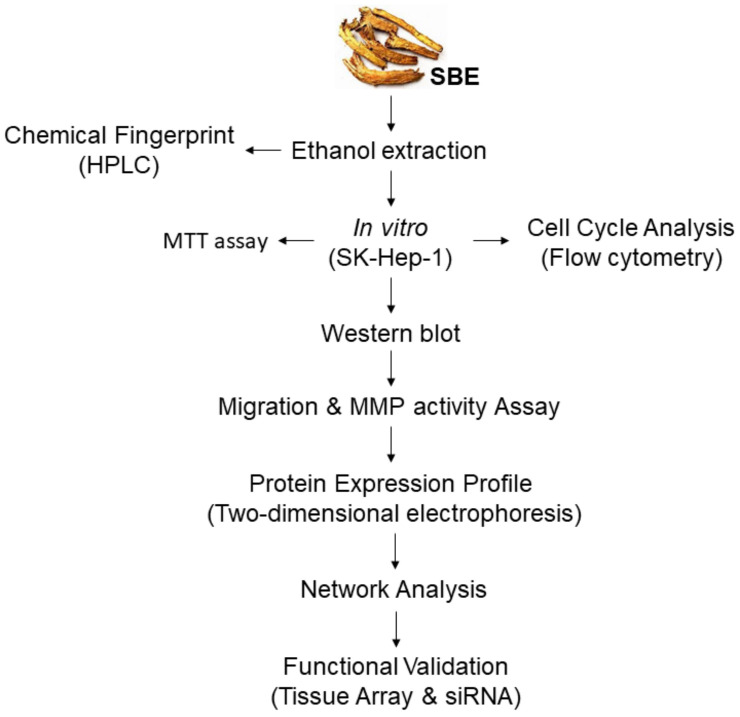
Schematic flowchart of experiments.

**Figure 7 ijms-25-03073-f007:**
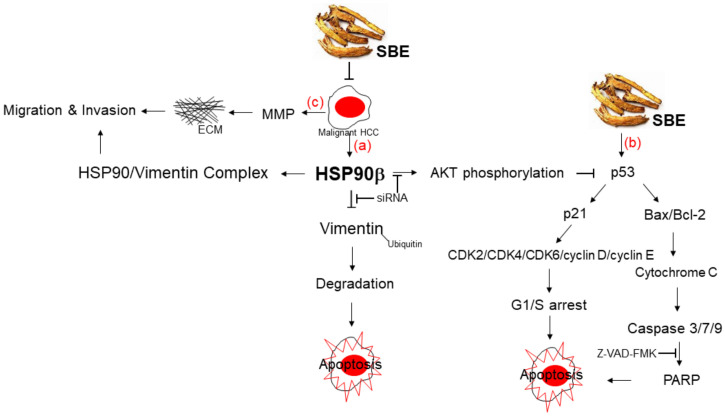
Schematic diagram of SBE-mediated G1/S arrest and apoptosis through regulating AKT phosphorylation and p53-bax-caspase-3/7/9-cytochrome c as well as EMT modulation. HSP90β plays a central role in mediating degradation of vimentin and dysregulation of cyclin-CDK, eventually exerting the inhibitory function in HCC metastasis.

**Table 1 ijms-25-03073-t001:** List of identified protein spots.

Spot No.	Protein Name	Accession Number	Score (Coverage) ^1^	Mw/pI	Function
1	ATPase subunit d	O75947	61 (50%)	18.41/5.22	Mitochondrial membrane ATP synthase produces ATP from ADP in the presence of a proton gradient across the membrane which is generated by the electron transport complexes of the respiratory chain.
2	Rho GDI 1	P52565	88 (50%)	20.57/6.73	Controls Rho proteins homeostasis. Through the modulation of Rho proteins, may play a role in cell motility regulation.
3	TCTP	P13693			Involved in calcium binding and microtubule stabilization.
4	Proteasome subunit alpha type-5	P28066	98 (58%)	26.56/4.74	The 26S proteasome plays a key role in the maintenance of protein homeostasis by removing misfolded or damaged proteins that could impair cellular functions, and by removing proteins whose functions are no longer required.
5	Tropomyosin alpha-3 chain	P06753	99 (45%)	27.39/4.77	Binds to actin filaments in muscle and non-muscle cells.
6	14-3-3 protein sigma	P31947	86 (54%)	27.87/4.68	An adapter protein implicated in the regulation of a large spectrum of both general and specialized signaling pathways.
7	Peroxiredoxin-6	P30041	197 (82%)	25.02/6.02	A thiol-specific peroxidase that catalyzes the reduction of hydrogen peroxide and organic hydroperoxides to water and alcohols, respectively.
8	Heat shock 27 kDa protein	P04792	99 (47%)	22.83/5.98	Small heat shock protein which functions as a molecular chaperone probably maintaining denatured proteins in a folding-competent state.
9	Nucleolar phosphoprotein B23	P06748	70 (31%)	29.62/4.47	Involved in diverse cellular processes such as ribosome biogenesis, centrosome duplication, protein chaperoning, histone assembly, cell proliferation, and regulation of tumor suppressors p53/*TP53* and ARF.
10	60S ribosomal protein L10E	P05388	82 (45%)	34.42/5.42	Ribosomal protein P0 is the functional equivalent of E.coli protein L10.
11	40S ribosomal protein SA	P08865	126 (51%)	31.88/4.84	Required for the assembly and/or stability of the 40S ribosomal subunit. Plays a role in cell adhesion to the basement membrane and in the consequent activation of signaling transduction pathways.
12	Vimentin (Fragment)	P08670	173 (66%)	53.55/5.06	Vimentins are class-III intermediate filaments found in various non-epithelial cells, especially mesenchymal cells.
13	26S proteasome regulatory subunit 6B	P43686	60 (16%)	47.45/5.09	A component of the 26S proteasome, a multiprotein complex involved in the ATP-dependent degradation of ubiquitinated proteins. Therefore, the proteasome participates in numerous cellular processes, including cell cycle progression, apoptosis, or DNA damage repair.
14	Vimentin	P08670	173 (66%)	53.55/5.06	Vimentins are class-III intermediate filaments found in various non-epithelial cells, especially mesenchymal cells.
15	75 kDa glucose-regulated protein	P38646	62 (20%)	74.02/5.97	A chaperone protein which plays an important role in mitochondrial iron-sulfur cluster (ISC) biogenesis. May play a role in the control of cell proliferation and cellular aging.
16	Heat shock cognate 71 kDa protein	P11142	178 (53%)	71.06/5.37	A molecular chaperone implicated in a wide variety of cellular processes, including protection of the proteome from stress, folding and transport of newly synthesized polypeptides, activation of proteolysis of misfolded proteins and the formation and dissociation of protein complexes.
17	60 kDa heat shock protein	P10809	122 (50%)	61.35/5.70	A chaperonin implicated in mitochondrial protein import and macromolecular assembly.
18	Tubulin alpha-1A chain	P68363	107 (51%)	50.48/5.01	Tubulin is the major constituent of microtubules, a cylinder consisting of laterally associated linear protofilaments composed of α- and β-tubulin heterodimers.
19	Tubulin beta chain	P07437	227 (52%)	50.09/4.75	Microtubules grow by the addition of GTP-tubulin dimers to the microtubule end, where a stabilizing cap forms.
20	Heat shock protein 90 kDa beta member 1	P14625	217 (52%)	95.3/4.75	A molecular chaperone that functions in the processing and transport of secreted proteins. Functions in endoplasmic reticulum associated degradation (ERAD).
21	type II cytoskeletal 8	P05787	197 (61%)	53.53/5.52	Together with KRT19, helps to link the contractile apparatus to dystrophin at the costameres of striated muscle.
22	Protein disulfide-isomerase A3	P30101	185 (45%)	57.15/5.98	A disulfide isomerase which catalyzes the formation, isomerization, and reduction or oxidation of disulfide bonds.
23	Calreticulin	P27797	120 (38%)	47.06/4.30	A calcium-binding chaperone that promotes folding, oligomeric assembly, and quality control in the endoplasmic reticulum via the calreticulin/calnexin cycle.
24	Glyceraldehyde-3-phosphate dehydrogenase	P04406	86 (57%)	36.20/8.26	Has both glyceraldehyde-3-phosphate dehydrogenase and nitrosylase activities, thereby playing a role in glycolysis and nuclear functions, respectively.
25	Triosephosphate isomerase	P60174	150 (76%)	26.81/6.51	Triosephosphate isomerase is an extremely efficient metabolic enzyme that catalyzes the interconversion between dihydroxyacetone phosphate and D-glyceraldehyde-3-phosphate in glycolysis and gluconeogenesis.

^1^ SwissProt 2021_03 (565,254 sequences; 203,850,821 residues).

## Data Availability

Data is contained within the article and Appendix A.

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
