# Peer review of "Scutellaria baicalensis Induces Cell Apoptosis and Elicits Mesenchymal–Epithelial Transition to Alleviate Metastatic Hepatocellular Carcinoma via Modulating HSP90β"

_ijms, 2024, doi:10.3390/ijms25053073_

Round 1
Reviewer 1 Report
Comments and Suggestions for Authors
The english could be improved at some points. Further information can be found within the review report. Overall, the quality of english is fine.
Author Response
Responses to Reviewer #1
- Line 48: Thanks very much for the advice. We have provided the reference [4], which indicated that patients with advanced stage of HCC do not respond well to current therapy.
- Line 53: Thanks for the comments. Since advanced HCC is a complicated and systematic disease, treatment focusing on single factor such as sorafenib shows invalid and severe side-effects during HCC therapy. In this regard, we warrant the roles of baicalensis in alternative and complementary intervention for long-term HCC therapy. We have added the reference [8].
- Reference7-10: We have replaced the review references with original articles [10,11]. Thanks.
- Line 57/58: Thanks very much. We have provided the reference [13].
- Line 59: We have amended the error.
- Line 59-61: Thanks. We provided the reference [14] to certificate the efficacy of the bioactive components extracted from baicalensis.
- Line 72-74: Thanks very much. We have provided the reference [18] to prove the functional role of HSP90.
- Reference 14/15: We have replaced the review references with original articles [19,20].
- Line 76/77: Thanks very much. We have provided the reference [21] to verify the statement.
- 1A: According to your comment, we have added the statistics.
- 1B: We have reformed the panel description according to your suggestion to avoid ambiguity. Thanks.
- Fig 2A, 2B and 2C: We have added the statistics for all the comparisons. Thanks for the comment.
- 2D: We have rearranged the Fig. 2D according to your suggestion and added the description of method used in cell cycle analysis in the figure legend.
- Line 164: Yes. We have removed “of”.
- Line 165: Thanks for your comment. We have rewritten the phrase to avoid the misunderstanding.
- 3A: We have added the statistics. Thanks.
- 3B: 250 mg/mL is a working dosage to perform inhibitory effect of S. baicalensis in HCC cells rather suppress the migration of HCC cells by cytotoxicity. We have added the statement in the line 166 of the result section.
- 5B: Thanks for the suggestion and we have added the bars and statistics for better comparison in Hsp90b level between treatments of siRNAs and SBE.
- 5C: Yes. We have added the color scheme description according to your comment.
Reviewer 2 Report
Comments and Suggestions for Authors
The article Scutellaria baicalensis induces cell apoptosis and elicits mesenchymal-epithelial transition to alleviate metastatic hepatocellular carcinoma via modulating Hsp90beta presents compelling and highly relevant results in the medical field. However, it requires improvements and I would like to make the following recommendations:
1. The Materials and Methods section should be added after the Introduction.
2. The Materials and Methods section should include a detailed flow chart outlining the study methodology.
3. The Discussion section is modest and requires more information from relevant literature, as well as addressing topics currently under active research. I recommend discussing the involvement of microbiota transfer in hepatocarcinoma and valuable information on this can be found in the article at https://doi.org/10.3390/biomedicines11112930. Additionally, a brief discussion on the involvement of microRNA in liver cancer, a topical subject addressed in https://doi.org/10.3390/biomedicines11072058, would be particularly interesting.
4. It is advisable to add a Conclusions section to summarize the obtained results.
Comments on the Quality of English LanguageMinor editing of English language required
Author Response
Responses to Reviewer
- Thanks for the suggestion, the articles are arranged according to the publisher's regulations. The “Materials and Methods” section is added after “Discussion” section, which is followed by the author’s guideline.
- We have added the detailed flow chart of study methodology in the 5.1 of the ‘Materials and Methods” section to particularly indicate the goals and approaches of current study.
- We agree with your constructive suggestion and the relationship between gut microbiota as well as RNA molecules and HCC progression is included in the Discussion section [line 350].
- As suggested, we have added “Conclusions” to summarize our findings in the current study [line 361].
Reviewer 3 Report
Comments and Suggestions for Authors
In this manuscript, the authors investigated the therapeutic effects of Scutellaria baicalensis extract (SBE) on hepatocellular carcinoma cell line (SK-Hep-1). They demonstrated that SBE causes several anti-cancerous effects, including apoptosis, cell cycle arrest, and mesenchymal-epithelial transition (MET). Moreover, using a proteomics approach, the authors found that SBE suppressed the expression of HSP90β protein. These findings are interesting, but quite preliminary. There are some serious issues as follows:
1. There are many studies demonstrating that SBE and its components has potent anti-cancerous effects. Therefore, the results shown in Figures 1-3 should be moved to supplementary figures.
2. At least, one more cell line should be used for all experiments. It should be verified that the observations in SK-Hep-1 cells are replicated in another cell line.
3. In order to conclude that the suppression of HSP90β has an important role in the mechanism of SBE's action, the authors must demonstrate that the exogenous overexpression of HSP90β abrogates the anti-cancerous effects of SBE.
4. Figure 4D: Dysregulation of the network should be verified in vitro.
5. Figure 5A: Statistical analysis should be performed. In particular, it should be determined whether the expression of HSP90β is associated with the malignancy of HCC.
6. Figure 5C: In general for the ubiquitination assay, immunoprecipitation precedes SDS-PAGE. Detail information is missing in legend and the materials and methods.
7. It should be described elsewhere that Sk-Hep-1 has wild type TP53 gene.
Author Response
Responses to Reviewer
- Thanks for the comment. Our manuscript is the first study to reveal the pharmacological effects and molecular mechanisms of SBE on malignant liver cancer cells. Figure 1-3 are the results indicating the responses of normal hepatic cells as well as SK-Hep-1 cells to SBE, which is the foundation for the further experiments. Therefore, we would tend to retain these figures.
- Following your comment, we applied SBE on both HepG2 and SK-Hep-1 cells. As expected, SBE application significantly inhibited the cell growth in HepG2 and SK-Hep-1 cells, demonstrating that SBE treatment should be effective in mitigating HCC progression. In this regard, SBE should exhibit similar effects in different hepatic cancer cells. Thanks for your constructive suggestion.
- The poorly differentiated SK-Hep-1 cells and higher grades of HCC patients show great amount of Hsp90b, which is a potent target for HCC therapy according to our results. Since SBE could perform the advantage aimed in multi-targets due to its property of multiple components, the synergistic effect with Hsp90b siRNAs also demonstrated the better efficacy in HCC treatment than that of individual medicine. Therefore, the anti-HCC ability of SBE would still show expected therapeutic outcome as exogenously overexpressed Hsp90b.
- 4D: Thanks for the comment. We have verified the results derived from the network analysis with Hsp90b siRNAs and the findings were indicated in the Fig. 5B, which implied that abrogation of Hsp90b would downregulate the vimentin expression and in turn arrest the HCC progression.
- Thanks for the constructive suggest and the statistical analysis of the 50 HCC patients and 9 normal subjects were performed as indicated in the Supplement Figure 1, which clearly indicates that high level of Hsp90b is positively associated with the malignancy of HCC.
- 5C: Detailed information of ubiquitination experiment is added in the figure legend and Material and Methods [5.4 Western blot analysis]. Thanks.
- The references [PMID: 32210718 & 34806333] have indicated that SK-Hep-1 cell has wild type of TP53 gene. Thanks for the comment.

Round 2
Reviewer 2 Report
Comments and Suggestions for Authors
The authors have incorporated the requested modifications and the article can be published in its present form.
Comments on the Quality of English LanguageMinor editing of English language required
Author Response
- Thanks for your comment.
-
The English editing has been modified by a native English speaker and the certification is attached as listed in the attached file.

Reviewer 3 Report
Comments and Suggestions for Authors
1. It has been well established that SBE has suppressive effects on cancer cell growth. It must be demonstrated that SBE also induces the effect via the same mechanism in HepG2 cells as observed in Sk-Hep-1 cells.
2. Given that the downregulation of Hsp90b plays a critical role in the mechanism of SBE's action, the cells exogenously overexpressing Hsp90b would retain cellular Hsp90b level to a certain extent even in the presence of SBE, and acquire resistance against SBE. These are absolutely essential data to draw the conclusion. The knockdown of Hsp90b is scientifically unnecessary to prove it.
3. It is still hard to understand why the authors performed immunoprecipitation and immunoblot with a ubiquitin antibody. Where is ubiquitinated vimentin?
Author Response
Responses to Reviewer
- Based on our results, SBE application significantly inhibited the cell growth in HepG2 and SK-Hep-1 cells, demonstrating that SBE treatment should be effective in suppressing HCC cell growth. In the current study, we mainly focus on the roles of SBE in alleviating the advanced HCC; therefore, the detailed mechanisms of SBE involved in arresting poorly differentiated SK-Hep-1 cell rather other HCC cells are thoroughly investigated. Since the differentiated grades are quite divergent for HepG2 and SK-Hep-1 cells, the molecular cascades under SBE treatment would not be exactly the same among various HCC cells. However, SBE could perform the advantage in activating on multi-targets; we believe the inhibitory effects and underlying mechanisms of SBE in HepG2 cells should be at least partially similar with that in the SK-Hep-1 cells. Thanks for your constructive suggestion.
- According to previous reports, Hsp90β would inhibit the STUB1-induced degradation of YTHDF2 to elevate the expression of YTHDF2 and to further boost the drug resistance of HCC (pmid: 37515378, 31659110). Herein, our results deepen the understanding of how SBE is applied to mitigate HCC progression and provide potential targets for treating HCC; the drug resistance of SBE will be investigated in future study. Thanks for the comment.
- We have demonstrated the functional roles of vimentin in HCC metastasis and demonstrated that vimentin is a feasible marker for EMT of HCC cells (pmid: 22387118). In addition, the degradation of vimentin is closely linked to the inhibitory efficacy of SBE in SK-Hep-1 cells as implied in the network analysis. In this regard, we evaluated the degraded forms of vimentin caused by ubiquitination as shown in the Fig. 5C where the ubiquitinated vimentin fragments were identified with vimentin antibody after IP with ubiquitin antibody. Thanks.
Round 3
Reviewer 3 Report
Comments and Suggestions for Authors
1. As the reviewer indicated, the anti-cancerous effects of SBE has been demonstrated in various cell lines, so that the authors must show something new other than the inhibition of cancer cell growth. Although the authors showed new mechanism of action of SBE in one cell line, the new finding cannot be conclusive without verifying it in another cell line.
2. Because HSP90b is the most important protein in the present study and the authors concluded that SBE suppress a HCC cell line by modulating HSP90b as seen in the title, the cause-and effect relationship between SBE's anti-cancerous effect and HSP90b downregulation must be clarified.
3. The experimental procedures for the ubiquitination assay should be added. In general, proteasome degrads target proteins into oligopeptides with 2-8 amino acid. Such fragmentation of vimentin shown in Fig 5C could arise from other reasons.
Author Response
Thanks for your comments for improving the quality of our manuscript.